# HUMAN-AI COORDINATION VIA HUMAN-REGULARIZED SEARCH AND LEARNING

## ABSTRACT

We consider the problem of making AI agents that collaborate well with humans in partially observable fully cooperative environments given datasets of human behavior. Inspired by piKL, a human-data-regularized search method that improves upon a behavioral cloning policy without diverging far away from it, we develop a three-step algorithm that achieve strong performance in coordinating with real humans in the Hanabi benchmark. We first use a regularized search algorithm and behavioral cloning to produce a better human model that captures diverse skill levels. Then, we integrate the policy regularization idea into reinforcement learning to train a human-like best response to the human model. Finally, we apply regularized search on top of the best response policy at test time to handle out-of-distribution challenges when playing with humans. We evaluate our method in two large scale experiments with humans. First, we show that our method outperforms experts when playing with a group of diverse human players in ad-hoc teams. Second, we show that our method beats a vanilla best response to behavioral cloning baseline by having experts play repeatedly with the two agents.

## 1 INTRODUCTION

One of the most fundamental goals of artificial intelligence research, especially multi-agent research, is to produce agents that can successfully collaborate with humans to achieve common goals. Although search and reinforcement learning (RL) from scratch without human knowledge have achieved impressive superhuman performance in competitive games (Silver et al., 2017; Brown & Sandholm, 2019), prior works (Hu et al., 2020; Carroll et al., 2019) have shown that agents produced by vanilla multi-agent reinforcement learning do not collaborate well with humans.

A canonical way to obtain agents that collaborate well with humans is to first use behavioral cloning (BC) (Bain & Sammut, 1996) to train a policy that mimics human behavior and then use RL to train a best response (BR policy) to the fixed BC policy. However, such an approach has a few issues. The BC policy is hardly a perfect representation of human play. It may struggle to mimic strong players' performance without search (Jacob et al., 2022). The BC policy's response to new conventions developed during BR training is also not well defined. Therefore, the BR policy may develop strategies that exploit those undefined behaviors and confuse humans and causes humans to diverge from routine behaviors or even quit the task because they believe the partner is non-sensible.

Recently, Jacob et al. (2022) introduced piKL, a search technique regularized towards BC policies learned from human data that can produce strong yet human-like policies. In some environments, the regularized search, with the proper amount of regularization, achieves better performance while maintaining or even improving its accuracy when predicting human actions. Inspired by piKL, we propose a three-step algorithm to create agents that can collaborate well with humans in complex partially observable environments. In the first step, we repeatedly apply imitation learning and piKL (piKL-IL) with multiple regularization coefficients to model human players of different skill levels. Secondly, we integrate the regularization idea with RL to train a human-like best response agent (piKL-BR) to the agents from step one. Thirdly and finally, at test time, we apply piKL on the trained best response agent to further improve performance. We call our method piKL3.

We test our method on the challenging benchmark Hanabi (Bard et al., 2020) through large-scale experiments with real human players. We first show that it outperforms human experts when partnering

with a group of unknown human players in an ad hoc setting without prior communication or warm-up games. Players were recruited from a diverse player group and have different skill levels. We then evaluate piKL3 when partnered with *expert* human partners. We find that piKL3 outperforms an RL best response to a behavioral cloning policy (BR-BC) – a strong and established baseline for cooperative agents – in this setting.

## 2 Related Work

The research on learning to collaborate with humans can be roughly categorized into two groups based on whether or not they rely on human data. With human data, the most straightforward method is behavioral cloning, which uses supervised learning to predict human moves and executes the move with the highest predicted probability. The datasets often contain sub-optimal decisions and mistakes made by humans and behavioral cloning inevitably suffers by training on such data. A few methods from the imitation learning and offline RL community have been proposed to address such issues. For example, conditioning the policy on a reward target (Kumar et al., 2019; Chen et al., 2021) can help guide the policy towards imitating the human behaviors that achieve the maximum future rewards at test time. Behavioral cloning with neural networks alone may struggle to model sufficiently strong humans, especially in complex games that require long-term planning (McIlroy-Young et al., 2020). Jacob et al. (2022) address this issue by regularizing search towards a behavioral cloning policy. The proposed method, piKL, not only improves the overall performance as most search methods do, but also achieves better accuracy when predicting human moves in a wide variety of games compared to the behavioral cloning policy on which it is based.

Human data can also be used in combination with reinforcement learning. Observationally Augmented Self-Play (OSP) (Lerer & Peysakhovich, 2019) augments the normal MARL training procedure with a behavioral cloning loss on a limited amount of data collected from a test time agent to find an equilibrium policy that may work well with that agent. OSP increases the probability of learning conventions that are compatible with the test time agents. However it may not be able to model partners with diverse skill levels given a large aggregation of data from various players. We can also use RL to train a best response policy to the behavioral cloning policy (Carroll et al., 2019). This method is the optimal solution given a perfect human model. In practice, however, RL is prone to overfitting to the imperfections of the human model. In addition, RL alone may not be sufficient in practice to learn superhuman strategies (Silver et al., 2018; Brown & Sandholm, 2019).

A parallel research direction seeks to achieve better human-AI coordination without using any human data. Some of them take inspiration from human behavior or the human learning process. Hu et al. (2020) enforce the RL policies to not break the symmetries in the game arbitrarily, a common practice of human players in certain games. Inspired by humans' learning and reasoning process, Off-belief learning (Hu et al., 2021c) and K-level reasoning (Costa-Gomes & Crawford, 2006; Cui et al., 2021b) train sequences of policies with increasing cognitive capabilities. Both methods achieve strong performance with a human proxy model trained with behavioral cloning. Another group of methods use population-based training and various diversity metrics (Strouse et al., 2021; Lupu et al., 2021; Tang et al., 2021) to first obtain a set of different policies and then train a common best response that may generalize better to human partners than best response to a single RL policy.

## 3 Background

### 3.1 Dec-POMDP and Deep Reinforcement Learning

We consider human-AI coordination in a decentralized partially observable Markov decision process (Dec-POMDP) (Nayyar et al., 2013). A Dec-POMDP consists of $N$ agents indexed by $(1, \ldots, N)$, a state space $\mathcal{S}$, a joint action space $\mathcal{A} = \mathcal{A}^1 \times \cdots \times \mathcal{A}^N$, a transition function $\mathcal{T} : \mathcal{S} \times \mathcal{A} \to \mathcal{S}$, a reward function $r : \mathcal{S} \times \mathcal{A} \to \mathbb{R}$ and a set of observation function $o^i = \Omega^i(s)$, $s \in \mathcal{S}$ for each agent $i$. We further assume that the joint actions $\mathbf{a}$ and rewards $r$ are observable by all agents. We then define the trajectory of true states until time step $t$ as $\tau_t = (s_0, \mathbf{a}_0, r_0, \ldots, s_t)$ and its partially observed counterpart (action-observation history, AOH) for agent $i$ as $\tau_t^i = (o_0^i, \mathbf{a}_0, r_0, \ldots, o_t^i)$. An agent's policy $\pi^i(\tau_t^i) = P(a_t^i | \tau_t^i)$ maps each possible AOH to a distribution over the action space of

that agent. We use $\boldsymbol{\pi}$ to denote the joint policy of all agents and $\boldsymbol{\pi}^{-i}$ to denote the joint policy of all other agents excluding agent $i$.

Deep multi-agent RL (MARL) has been successfully applied in many Dec-POMDP environments. Deep MARL algorithms often consist of a strong RL backbone such as (recurrent) DQN (Kapturowski et al., 2019) or PPO (Schulman et al., 2017) and additional modules such as a centralized value function (Yu et al., 2021), value-decomposition (Sunehag et al., 2017; Rashid et al., 2018) to handle challenges posed by having multiple agents. In this paper, we use recurrent DQN to train a best response to fixed policies. Specifically, a recurrent network is trained to model the expected total return for each action given the input AOH, $Q(\tau_t^i, a) = \mathbb{E}_{\tau \sim P(\tau_t | \tau_t^i)} R(\tau_t)$ where $R(\tau_t) = \sum_{t' \geq t} \gamma^{(t'-t)} r_t$ is the sum of discounted future reward by unrolling the joint policy $\boldsymbol{\pi}$ on the sampled true game trajectory until termination. The joint policy is the greedy policy derived from each agent's Q-function.

## 3.2 SEARCH AND REGULARIZED SEARCH IN DEC-POMDP

Search has been critical to achieve superhuman performance in many games (Silver et al., 2018; Brown & Sandholm, 2019; Bakhtin et al., 2021). SPARTA (Lerer et al., 2020) and its faster and more generalized variant Learned Belief Search (Hu et al., 2021b) are competitive and efficient search algorithms in Dec-POMDPs. SPARTA assumes that a joint blueprint policy (BP) $\boldsymbol{\pi}$ has been agreed on beforehand. In single-agent SPARTA, one agent performs search at every time step assuming that their partners follow the BP. Specifically, the search agent $i$ keeps track of the belief $\mathcal{B}(\tau_t^i) = P(\tau_t | \tau_t^i, \boldsymbol{\pi}^{-1})$, which is the distribution of the trajectory of states given the AOH and partners' policies. It picks the action $a'$ that returns the highest sum of undiscounted future rewards assuming $a'$ is executed at time $t$ and everyone follows the joint BP afterwards, i.e.

$$a_t^i = \arg\max_a Q_{\boldsymbol{\pi}}(\tau_t^i, a) = \arg\max_a \mathbb{E}_{\tau_t \sim \mathcal{B}(\tau_t^i)}[r(\tau_t, a) + R_{\boldsymbol{\pi}}(\tau_{t+1})], \qquad (1)$$

where $r(\tau_t, a)$ is the reward at time $t$ after executing $a$ and $R_{\boldsymbol{\pi}}(\tau_{t+1})$ is the sum of future rewards following joint policy $\boldsymbol{\pi}$. This notation assumes a deterministic transition function as the randomness can be absorbed into the belief function.

The belief tracking in SPARTA is computationally expensive and may have null support when partners deviate even slightly from the BP. Learned Belief Search (Hu et al., 2021a, LBS) mitigates those problems by using a neural network belief model $\hat{\mathcal{B}}$ trained with data generated by the BP with some exploration. For environments where the observation can be factorized into public and private parts, such as Hanabi, LBS also proposes to use a two-stream architecture where one stream with LSTM takes the public information as input while the other stream consists of only feed-forward layers takes the private information. The outputs of the two streams are fused to compute Q-values. This special architecture further reduces the computation cost as it no longer needs to re-unroll the LSTM from the beginning of the game for each sampled $\tau_t$.

Although the learned belief technique was originally proposed to speed up SPARTA, it becomes critical in scenarios where the game trajectory does not exactly follow the assumed joint policy. For example, when playing with humans, humans' real moves may differ from our model's predictions and the learned belief model can often generalize well in those cases. In this paper we use LBS as the search component in piKL.

Jacob et al. (2022) show that the output policy of search algorithms can diverge quite far from the underlying blueprint policy used for rollouts and value estimation, which is undesirable in environments where collaborating with human partners is crucial. In general, they propose to sample actions following

$$P(a) \propto \pi_{\text{anc}}(a|\tau_t^i) \cdot \exp\left[\frac{Q_{\pi_{\text{roll}}}(\tau_t^i, a)}{\lambda}\right], \qquad (2)$$

where $Q_{\pi_{\text{roll}}}(\tau_t^i, a)$ is the value output of a search algorithm like SPARTA, Monte Carlo tree search or regret matching using $\pi_{\text{roll}}$ as the blueprint policy for rollouts and/or value estimation, $\pi_{\text{anc}}(a|\tau_t^i)$ is the anchor policy that we want our final policy to be close to and $\lambda$ is a hyper-parameter controlling the degree of regularization. In fully cooperative games where mixed strategies are not necessary, Jacob et al. (2022) show that a greedy variant that select the $\arg\max$ works better in practice.

## 4 METHOD

In this section we introduce piKL3. We first use piKL-IL with a probability distribution over the regularization parameter $\lambda$ to model human players with varying skill levels. Then, we use RL regularized toward the behavioral cloning policy to train a human-compatible best response piKL-BR. Finally, we use piKL-LBS at test time with high regularization toward the BR to fix severe mistakes when playing with real humans.

### 4.1 PIKL-IL FOR MODELING DIFFERENT LEVELS OF HUMAN PLAY

---

**Algorithm 1** piKL-IL: modeling human with different skill levels. $P(\lambda)$ can be a discrete uniform distribution over a set of values or over a set of Gaussian distributions centered around those values. piKL-LBS($\lambda_i$, $\pi_{\text{roll}}$, $\hat{\mathcal{B}}$, $\pi_{\text{anc}}$) is a function to act following Eq. 2 or its greedy variant. It samples from the learned approximate belief model $\tau_t \sim \hat{\mathcal{B}}(\tau_t^i)$ to estimate $Q_{\pi_{\text{roll}}}$.

---

1: **procedure** PIKL-IL($\pi_{BC}$, $P(\lambda)$, $k$, $d$)
  $\triangleright \pi_{BC}$: behavioral cloning policy trained from human data;
  $\triangleright P(\lambda)$ : distribution of $\lambda$ ;
  $\triangleright k$: number of iterations
  $\triangleright d$: size of the dataset
2: $\quad$ $\pi_{\text{piKL-IL}} \leftarrow \pi_{BC}$
3: $\quad$ **for** $i \leftarrow 1, \ldots, k$ **do**
4: $\quad\quad$ Train a belief model $\hat{\mathcal{B}}$ from self-play games of $\pi_{\text{piKL-IL}}$
5: $\quad\quad$ Initialize dataset $\mathcal{D} = \emptyset$
6: $\quad\quad$ **while** len($\mathcal{D}$) $< d$ **do**
7: $\quad\quad\quad$ Sample $\lambda_i \sim P(\lambda)$ for every player independently
8: $\quad\quad\quad$ Generate a game $g$ where player $i$ follows piKL-LBS($\lambda_i$, $\pi_{\text{piKL-IL}}$, $\hat{\mathcal{B}}$, $\pi_{BC}$)
9: $\quad\quad\quad$ Add the game $g$ to dataset $\mathcal{D}$
10: $\quad\quad$ **end while**
11: $\quad\quad$ Train a new policy $\pi'$ with behavior cloning on $\mathcal{D}$
12: $\quad\quad$ $\pi_{\text{piKL-IL}} \leftarrow \pi'$
13: $\quad$ **end for**
14: $\quad$ **return** $\pi_{\text{piKL-IL}}$
15: **end procedure**

---

PiKL-IL is a search-augmented imitation learning method. It first trains an imitation policy $\pi_{BC}$ via behavioral cloning (Bain & Sammut, 1996) on a dataset collected from the population of humans we want to model. Then piKL-IL iteratively improves a policy $\pi_{\text{piKL-IL}}$, alternating between generating higher quality data with piKL-LBS and training a better model using the generated dataset to produce a new $\pi_{\text{piKL-IL}}$. Each iteration of piKL-LBS uses $\pi_{BC}$ as the anchor policy $\pi_{\text{anc}}$ and $\pi_{\text{piKL-IL}}$ as the rollout policy $\pi_{\text{roll}}$ in Eq. 2, so that we always anchor the generated data to never differ too much from human play, while using the best rollout policy so far to generate the next. The pseudocode is in Algorithm 1.

piKL was shown to maintain the same or even higher prediction accuracy on human moves while achieving much higher performance with certain $\lambda$s, indicating that it may be better at modeling stronger human players (Jacob et al., 2022) than behavioral cloning. As $\lambda$ becomes smaller, the prediction accuracy drops while the performance keeps increasing, moving closer to the unregularized search policy. We therefore use a distribution of $\lambda$s to generate a spectrum of policies with strength ranging from average human players to exceptional policies that still resemble human behaviors reasonably well. When training a new policy on the generated data, we can condition the policy on the $\lambda$ so that we can explicitly control the distribution of different skill levels in the subsequent iterations as well as in the piKL-BR of Section 4.2.

Theoretically, we should apply multi-agent piKL-LBS but it is too computationally demanding to generate enough data for imitation learning. Instead we run single-agent piKL-LBS with learned beliefs independently for both players. Prior work (Jacob et al., 2022) shows that although running piKL-LBS independently for more than one player lacks theoretical guarantees because both players are unsoundly assuming that the other player is playing according to the pre-search policy when both

players in fact play according to the post-search policy, the algorithm still achieves high performance since all policies are regularized towards the same $\pi_{BC}$, thus the learned belief models are still in practice a good approximation of the true beliefs despite the shift in the underlying policies. For a theoretically sound version, we could alternatively run single-agent piKL-LBS on only one player and only collect training data only from that player's perspective while fixing the other player to only play the pre-search policy that the learned belief assumes they will.

Once we have collected enough data, we can train a new model with imitation learning and proceed to a new iteration. The process can be repeated until $\pi_{\text{piKL-IL}}$ stops improving. Note that the anchor policy is always the same human behavior-cloned policy to prevent the final policy from drifting away from human conventions.

The algorithm is presented here in the iterative form. However, if computational resources permit, it can be formulated as an asynchronous RL algorithm similar to AlphaZero (Silver et al., 2018) where $\pi_{\text{piKL-IL}}$ is constantly trained with data from a replay buffer while many parallel workers generate games with piKL-LBS and add them to the buffer.

## 4.2 PIKL-BR FOR A HUMAN-LIKE BEST RESPONSE

A popular approach to human-AI coordination is to train a best response to a human model. This BR training is similar to standard single-agent RL in POMDP settings as the partners are fixed policies and thus can be viewed as part of the environment. In practice, this method has a few issues due to the imperfection of the human model as well as the overfitting problem in RL.

Given a normally distributed dataset in which the majority of the humans have intermediate skill levels, the vanilla behavioral-cloning model often converges to an intermediate average score in self-play. Moreover, as observed in Jacob et al. (2022), BC often nontrivially underperforms even the average of the players it is trained on. This can make it hard for the BR agent to learn the true best response to stronger-than-average players, or even to average players. We can address this problem by training a BR $\pi_{\text{piKL-BR}}$ against the final piKL-IL policy $\pi_{\text{piKL-IL}}$ instead of the original human behavior cloning policy $\pi_{\text{BC}}$.

Similar to how single-agent RL can overfit to its exact training environment, an RL best response may overfit to its fixed neural partner, including finding unusual or out-of-distribution actions that happen to cause its partner to perform slightly better actions but that might not generalize to actual human players. In this case, instead of greedily picking an action that has slightly higher return as in normal RL, it would be better to err on the side of what humans tend to do in order to remain in-distribution. To address these issues, we propose to add piKL regularization during BR training.

Specifically, we train a policy $\pi_{\text{piKL-BR}}$ to be a best response to $\pi_{\text{piKL-IL}}$ via $Q$-learning, but we modify the $Q$-learning update as

$$Q(\tau_t^i, a_t) \leftarrow r_t(\tau_t, a) + \gamma \cdot Q(\tau_{t+1}^i, a_{t+1}'), \tag{3}$$

$$\text{where } a_{t+1}' = \arg\max_a [Q(\tau_{t+1}^i, a) + \lambda \cdot \log \pi_{\text{BC}}(\tau_{t+1}^i, a)], \tag{4}$$

and where the exploration policy is $\epsilon\text{-Greedy}[Q(\tau_t^i, a) + \lambda \cdot \log \pi_{\text{BC}}(\tau_t^i, a)]$.The difference from the normal Q-learning is highlighted in red. At test time, $\epsilon$ is set to $0$. The $\lambda$ here can be set to a smaller value than that in piKL-IL because the main purpose is no longer modeling human moves but rather regularization and tie-breaking when multiple actions have small differences in expected return.

It is worth noting that if we run piKL-IL with the same small $\lambda$ for *many* iterations, then the final $\pi_{\text{piKL-IL}}$ with the small $\lambda$ input converges to the same policy as piKL-BR. PiKL-BR is the amortized model-free version of piKL-IL and this step can be omitted if there are enough resources to run piKL-IL for enough iterations with additional $\lambda$s.

## 4.3 PIKL-LBS FOR ROBUSTNESS AGAINST OOD ERRORS

The $\pi_{\text{piKL-BR}}$ policy is a strong human-like policy that performs well with piKL-IL and humans who play similarly. When playing with a diverse group of human players in real life, however, it may suffer from out-of-distribution (OOD) errors when encountering trajectories that have low probability under training distributions. The actions produced by $\pi_{\text{piKL-BR}}$ on OOD input sequences can be arbitrarily bad as the neural network has never been trained on such data.

| $\lambda$ Input | 10 | 5 | 2 | 1 |
|---|---|---|---|---|
| Self-Play | $21.81 \pm 0.05$ | $22.06 \pm 0.04$ | $22.36 \pm 0.04$ | $22.66 \pm 0.03$ |
| w/ piKL-BR | $22.99 \pm 0.04$ | $23.10 \pm 0.03$ | $23.20 \pm 0.03$ | $23.34 \pm 0.03$ |

Table 1: The performance of piKL-IL in self-play and in cross-play with piKL-BR. The piKL-IL model is a single neural network model whose input contains the mean of Gaussian distribution from which the $\lambda$ of the player is sampled. It is evaluated with all possible input conditions ($\lambda$ input). Each cell (mean $\pm$ standard error) is evaluated over 5000 games with different random seeds. For reference, $\pi_{\text{BC}}$ learned on the original human dataset gets $19.72 \pm 0.10$ in self-play.

However, search or other model-based planning algorithms can mitigate this problem by avoiding the most devastating mistakes, because it often takes only a few steps of simulated rollouts to directly observe the negative outcomes caused by those mistakes. Inspired by this observation, we run piKL-LBS at test time on top of $\pi_{\text{piKL-BR}}$. We use $\pi_{\text{piKL-BR}}$ for *both* of $\pi_{\text{anc}}$ and $\pi_{\text{roll}}$ in Eq. 2 when it is our turn, and assume the partner acts according to $\pi_{\text{piKL-IL}}$ on their turn. The belief model is trained on data generated by cross-play between piKL-BR and piKL-IL. Since the main purpose of this step is to avoid catastrophic OOD errors that are usually associated with substantially lower $Q$-values, we can set $\lambda$ high so the search policy stays close to $\pi_{\text{piKL-BR}}$ in situations when the $Q$-values do not substantially differ.

## 5 EXPERIMENTAL SETUP

We implement and test our method in the Hanabi benchmark environment (Bard et al., 2020). In this section, we introduce the game rules of Hanabi, as well as how we implement piKL[3] and a best response to $\pi_{\text{BC}}$ (BR-BC) baseline.

Hanabi is a 2 to 5 player card game. In this paper we use the standard 2-player version. The deck consists of five color suits and each suit has ten cards divided into five ranks with three 1s, two 2s, two 3s, two 4s and one 5. At the beginning, each player draws five cards from the shuffled deck. Players can see other players' cards but not their own. On each turn, the active player can either hint a color or rank to another player or play or discard a card. Hinting a color or rank, will inform the recipient which cards in their hand have that specific color/rank. Hinting costs an information token and the team starts with eight tokens. The goal of the team to play exactly one card of each rank 1 to 5 of each color suit, in increasing order of rank, The order of plays between different color suits does not matter. A successful play scores the team one point while a failed play one costs one life. If all three lives are lost, the team will get 0 in this game, losing all collected points. The maximum score is 25 points. The team regains a hint token when a card is discarded or when a suit is finished (playing all 5 of a suit successfully). The player draws a new card after a play or discard move. Once the deck is exhausted, the game terminates after each player makes one more final move.

We acquire a dataset of roughly 240K 2-player Hanabi games from BoardGameArena[1] to train the human policy $\pi_h$. The dataset contains all the games played in a certain period on that online platform and we do not perform any filtering. The dataset is randomly split into a training set of 235K games, a validation set of 1K games and a test set of 4K games. The average score of games in the training set is $15.88$. The policy $\pi_\theta$ is parameterized by a Public-LSTM neural network (See Hu et al. (2021b) or Section 3.2). The policy is trained to minimize the cross-entropy loss $\mathcal{L}(\theta) = -\mathbb{E}_{\tau^i \sim \mathcal{D}} \sum_t \pi_\theta(a_t^i | \tau_t^i)$. Note that it treats the AOH of each player $\tau_i$ as a separate data point for training. Similar to prior works (Hu et al., 2021c), we apply color shuffling Hu et al. (2020) for data augmentation. Every time we sample $\tau_i \sim \mathcal{D}$, we generate a random permutation $f$ of the five colors, e.g. $f : a\,b\,c\,d\,e \to b\,d\,c\,a\,e$, and apply $f$ to both the input and the target of the training data. This model is trained with Adam (Kingma & Ba, 2014) optimizer until the prediction accuracy on the validation set peaks. The converged $\pi_h$ gets $19.72 \pm 0.10$ in self-play and $63.63\%$ in prediction accuracy on the test set. In evaluation, we take the $\arg\max$ from the policy instead of sampling, which also explains why it achieves higher average score than the training set.

PiKL-LBS requires a learned approximate belief model. In Hanabi, the belief model takes the same AOH $\tau^i$ as the policy and returns a distribution over player $i$'s own hand. The hand consists of 5

---
[1]en.boardgamearena.com

cards and we can predict them sequentially from the oldest to the newest based on the time they are drawn. The belief network $\phi$ consists of an LSTM encoder to encode sequence of observations and an LSTM decoder for predicting cards autoregressively. Note that the belief is a function of partner's policy. The belief for a given policy $\pi$ partnering with $\rho$ is trained with cross-entropy loss $\mathcal{L}(\phi) = -\mathbb{E}_{\tau^\pi \sim \mathcal{D}(\pi,\rho)} \sum_t \sum_j \log p_\phi(c_j | \tau_t^\pi, c_1, \cdots . c_{j-1})$, where $c_j$ is the $j$-th card in hand to predict. $\mathcal{D}(\pi, \rho)$ is an infinite data stream generated by cross-play using $\pi$ and $\rho$ and $\tau^\pi \sim \mathcal{D}$ means that we only use data from $\pi$'s perspective for training. In piKL-IL, we use $\pi = \rho = \pi_{\text{roll}}$.

We set $P(\lambda)$ to be a uniform mixture of truncated Gaussian distributions to model players of diverse skill levels. Specifically, we use Gaussian distributions $\mathcal{N}(\mu, \sigma^2)$ truncated at 0 and $2\mu$ with $(\mu, \sigma) = (1, 1/4), (2, 2/4), (5, 5/4), (10, 10/4)$ and each Gaussian is sampled with equal probability. We generate $d = 250K$ games in each iteration to train the new policy and we find that one outer iteration ($k = 1$ in Algo. 1) is sufficient to achieve good performance in Hanabi. In every LBS step, we perform $M = 10K$ Monte Carlo rollouts evenly distributed over $|\mathcal{A}|$ legal actions. We sample $M/|\mathcal{A}|$ valid private hands from the belief model to reset the simulator for rollouts. Invalid sampled hands are rejected. With this setting, each game with 2 player running piKL-LBS independently takes roughly 5 minutes with 1 GPU and we use 500 GPUs in parallel for 42 hours to generate the entire dataset. To better imitate policies under different $\lambda$s, we feed the $\mu$ of the Gaussian distribution from which the $\lambda$ is sampled to the policy network in the form of a one-hot vector concatenated with the input. The self-play performance of the piKL-IL model conditioning on different $\lambda$ input is shown in the top row of Table 1. Clearly, piKL-IL performs significantly better than $\pi_h$ and the score increases as regularization $\lambda$ decreases.

The BR is trained under a standard distributed RL setup where many parallel workers generate data with cross-play between the training policy and the fixed IL policy. The generated data is added into a prioritized replay buffer Schaul et al. (2015) and the training loop samples mini-batches of games to update the policy with TD errors. We use the same Public-LSTM architecture for the BR policy as it will also be used in piKL-LBS at test time. The BR policy explores with $\epsilon$-greedy to a distribution of $\epsilon$ sampled every new game while the IL policy does not explore but it samples a new $\lambda$ input from $\{1, 2, 5, 10\}$ every game. The $\lambda$ in Eq. 4 is set to 0.1. The cross-play performance between the converged piKL-BR and piKL-IL is shown in the bottom row of Table 1. As expected, piKL-BR is better at collaborating with piKL-IL than piKL-IL itself and the gap shrinks as the regularization $\lambda$ decreases. The reasons are that piKL-BR is trained with lower regularization and RL can optimize for multi-step best response while search can only optimize for one step.

We run piKL-LBS on the piKL-BR policy with high regularization $\lambda = 2$. The search assumes that our blueprint is $\pi_{\text{bp}}$ and our partner always follows $\pi_{\text{IL}}$, the final output policy of piKL-IL. To avoid predicting the $\lambda$ input for partner model $\pi_{\text{IL}}$, we replace it with an imitation learning policy $\pi'_{\text{IL}}$ trained on the same dataset as $\pi_{\text{IL}}$ but without the $\lambda$ input. The belief model is trained the same way as in piKL-IL but with $\pi = \pi_{\text{BR}}$ and $\rho = \pi'_{\text{IL}}$ in $\mathcal{D}(\pi, \rho)$. The number of Monte Carlo rollouts per step is reduced to 5K to make it faster and suitable for real world testing.

Finally, we train an unregularized $\lambda = 0$ best response to the vanilla behavioral clone policy $\pi_{\text{BC}}$ as our baseline. This agent achieves $23.19 \pm 0.03$ in cross-play with $\pi_{\text{BC}}$ in convergence. This score is quite high considering that its partner $\pi_{\text{BC}}$ is much worse than the piKL-IL policy, indicating that the unreguarlized BR may be overfitting to the imperfect human model.

## 6  RESULTS

We carry out two large scale experiments with real humans to evaluate piKL3. The first experiment focuses on ad-hoc team play with a diverse group of players without any prior communication (zero-shot). In the second experiment, we invite a group of expert players to play multiple games with piKL3 and the BR-BC baseline in alternating order to further differentiate the gap between them.

The experiments are hosted on our customized version of the open sourced Hanab.Live platform[2]. The modified platform disables chat, observe and replay functionalities. Additionally, participants cannot create games themselves nor invite others to form a team. All games are created automatically following the design of the experiments below. We send each player an instruction document of the platform to make them familiar with the UI in advance.

---

[2]`https://github.com/Hanabi-Live/hanabi-live`

|  | w/ Human Experts | w/ BR-BC | w/ piKL3 |
|---|---|---|---|
| All Testers (56) | 14.54 ± 1.47 | **16.73 ± 1.27** | **17.18 ± 1.28** |
| Newcomer (2) | 0.00 ± 0.00 | 0.00 ± 0.00 | **10.00 ± 7.07** |
| Beginner (17) | 9.12 ± 2.65 | **14.82 ± 2.42** | 14.47 ± 2.63 |
| Intermediate (23) | 14.57 ± 2.27 | **19.48 ± 1.64** | 18.52 ± 1.79 |
| Expert (14) | **23.14 ± 0.60** | 16.93 ± 2.41 | 19.29 ± 2.14 |

Table 2: The performance of different groups of players partnering with a group of testers. *Testers* are recruited from diverse sources to represent the general population with different skill levels and different conventions in Hanabi. Each *tester* is matched with one of the available human experts, one BR-BC baseline agent and one piKL3 agent in random order. The bottom 4 rows shows the results for each subgroup of testers. The number in parentheses after the group name is the headcount of that group. Each cell contains mean ± standard error.

|  | w/ BR-BC | w/ piKL3 |
|---|---|---|
| Experts | 21.21 ± 0.59 | **22.23 ± 0.52** |
|  | 22.52% ± 3.96% | **31.86% ± 4.38%** |

Table 3: The performance of experts playing with BR-BC baseline and piKL3 in alternating order repeatedly for a maximum of 20 games in total. The cells contain mean ± standard error (top row) averaged over 111 (BR-BC) and 113 (piKL3) valid games, and percentage of perfect games (bottom row).

## 6.1 AD-HOC TEAM PLAY

In the first experiment, we recruited players with different skill levels from diverse sources. We posted invitations on the board game Reddit, the forum of BoardGameArena, Twitter and Facebook Ads, as well as on two popular Discord channels where enthusiasts discuss conventions and organize tournaments. This group of players are referred to as *testers*. A $40 or $80 gift card was sent to the testers who successfully completed the required games to encourage participation. Meanwhile, we recruited a group of expert human players from a well-known Discord group to study how well humans can do when playing with unknown partners. The *experts* have all played Hanabi for more than 500 hours. The *experts* were paid proportional to the time they spend waiting for and playing with the *testers*. Each *tester* signed up for a 45-60 minute time slot. During their session, they were automatically matched with the BR-BC baseline agent, the piKL3 agent and one of the available experts in random order. Usernames of all players including AI agents were randomly decided, to maintain anonymity of the participants and of which players were the experts or the agents. Both AI agents sampled a sleep time $t$ proportional to the entropy of $\mathrm{softmax}(Q)$ and waited for at least $t$ seconds before sending the action. This further helped to hide the identity of AI agents and to mitigate the potential side effect that piKL3 and the BR-BC need different amounts of time to compute an action.

The results are shown in Table 2. From the overall result in the first row, we see that both BR-BC and piKL3 outperformed human experts in this task, indicating that playing with a diverse range of players in the zero-shot ad-hoc setting is challenging for humans. Still, it is worth noting that the 2-player no-variant version of Hanabi is not a particularly popular variant within the advanced Hanabi community and many people mostly play with people they know so that they can discuss and perfect strategies after games.

We let *testers* self-identify as one of four skill levels spanning from newcomer who has just learned the rules to expert who has played extensively. The results for each skill level group together with the number of players from each group are shown in the table as well. Although the standard errors are large due to the small number of games within each group, we can see a trend that the AI agents generally worked better with non-expert players, while experts have a clear lead when collaborating with other experts. This is likely because experts' behaviors are more predictable and the community has converged to a few well-known convention sets that are easy to identify for the experts who follow them closely in forums and discussion channels. It might also reflect the fact that our training data matches a more diverse pool of players with fewer experts in it, since those experts tend to play on sites other than BoardGameArena, which is our only source of training data.

The difference between BR-BC and piKL3 in this set of results is not significant given the standard errors. Unfortunately, it is challenging to collect more games in this setting due to both the difficulty to recruit enough people with good intent and the logistic overhead to manage anonymous human-human matches.

We hypothesized that piKL3 might work better with experts thanks to its abilities to model stronger human players and to be more robust against OOD errors. We design a different experiment to verify this in the next section.

## 6.2 Repeated Games with Experts

In this experiment, we recruited a group of expert players to play with the two AI agents in alternating order for a maximum of 20 games. Each expert started randomly with one of the agents and switched to the other one after each game. They were aware of the matching rule and which AI was in their current game. As an incentive to do well, the players were compensated with $0.6 for every point they got. A terminated game with all life token lost counted as 0 points.

We collected 111 games for the BR-BC agent and 113 games for the piKL3 agent. The numbers are slightly different for the two agents because some games terminated unexpectedly due to platform related technical reasons. The average score and the percentage of perfect games for each agent are shown in Table 3. A perfect game is where the team achieves full score. Although the improvement may seem small numerically, the mechanics of Hanabi makes it increasingly difficult to improve as the score gets closer to a perfect 25. In RL training, for example, learning a 20-point policy from scratch takes roughly the same amount of time and data as improving that policy to a 21-point one.

piKL3 outperformed the BR-BC in terms of both average score and percentage of perfect games. Although the statistical significance of both results is somewhat limited given the amount of data $p = 0.097$ and $p = 0.058$, respectively, both are strongly suggestive and consistent with our initial hypothesis. It is also encouraging to see that piKL3 can achieve more perfect games with the experts. Experienced human players are particularly excited about perfect games as they are often significantly harder than getting 23 or 24 in a given deck.

Due to the limitation on budget and the overhead of managing experiments with humans, we were unable to collect enough games to perform ablations for each component of piKL3, nor to include more baselines. In this experiment, we focused on demonstrating the effectiveness of this combination compared to the popular BR-BC baseline. In practice, researchers may use any combination of the three components of piKL3 based on the properties and challenges of their specific domains.

Most existing works targeting human-AI coordination in Hanabi (Hu et al., 2021c; Cui et al., 2021a) do not directly use human data. They get around 16 points with a similar but slightly stronger $\pi_{BC}$ while piKL3 and BR-BC get around 23. Therefore, despite being interesting, it will not be a fair comparison to include them. Previously, Hu et al. (2020) also tested their agent with human players. However, these results are not directly comparable as they use a different scoring method that keeps all the points when losing all life tokens. Additionally, the population of the human testers have a profound impact on the numerical results.

## 7 Conclusion and Future Work

We present piKL3, a three-step algorithm centered around the idea of regularizing search, imitation learning and reinforcement learning towards a policy learned from human data. We performed two large-scale experiments with human players in the Hanabi benchmark to demonstrate the effectiveness of this method and its superiority over the popular baseline of training an ordinary best response to a plain imitation policy. We also for the first time report human experts' perhaps surprisingly low performance on zero-shot ad-hoc team play with a diverse population.

The main limitation of this method is that it requires large amounts of data to learn the initial human policy used as anchor and blueprint in piKL search. Therefore, an interesting future direction is to extend this method onto more domains, especially ones with less high quality human data. Another direction would be to create personalized regularization that adapts to each individual player in repeated games in an attempt to better model them.

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
