# OpenReview forum: "Human-AI Coordination via Human-Regularized Search and Learning"
_ICLR.cc/2023/Conference — Submitted to ICLR 2023_

### Official Review · Reviewer_yAwz · 2022-10-24

**Confidence:** 3
**Correctness:** 3
**Technical Novelty And Significance:** 3
**Empirical Novelty And Significance:** 3
**Recommendation:** 8

**Clarity, Quality, Novelty And Reproducibility:**

Quality: great execution of a conceptually simple idea. Impressive empirical evaluation.
Clarity: great writing.
Originality: the problem setting is rarely studied and seems to be important for the future research of human-AI collaboration.

**Strength And Weaknesses:**

Overall I quite like the spirit of this paper — for human-machine collaboration, human data is inherently scarce, especially for methods that rely on massive simulation and trial-and-error imitation learning. Training a human-like policy from a fixed set of human demonstration to generate reactive response for RL learning seems to be a good strategy. The proposed method is also conceptually simple — use a constrained BC method to plug “holes” in the human policy to prevent the RL learner from exploiting these errors to develop un-human behaviors. Finally, the scale of human study is laudable.

I don’t have major concerns. A few minor ones:
- Does \lambda truly capture the level of expertise in eq.2? Intuitively it’s an interpolation between “optimal play” and “human-like” play instead of different expertise levels. Besides, how much does training the policy against these policies matter in practice? There doesn’t seem to be an ablation study on this.
-  The general recipe for learning human-like behavior policy to train RL collaborators seems to be general and can be applied to other domains. The authors also seem to agree with this point in the “future work” section. The authors should discuss which aspect of the proposed method needs to be adapted in order to accommodate new domains.
- A piece of related work that’s missing from the paper is  Wang et al. [1]. They explored a similar venue of idea, including elements such as learning a human-like behavior policy to train RL collaborators and modeling diverse collaboration strategies, albeit in a robotics domain. It’d be great if the authors could compare and contrast their work with [1].

[1] Co-GAIL: Learning Diverse Strategies for Human-Robot Collaboration, Wang et al., 2021


**Summary Of The Paper:**

The paper presents a framework for training RL agents to collaborate with human players in 2-player Hanabi games. The key challenge of training such a policy is that human data is finite and fixed, which means that there needs to be a human-like behavior policy for the RL to learn to collaborate with. Prior work shows that such human-like policy trained with naive behavior cloning may make mistakes that can be exploited by the RL agent, resulting in the RL agents exhibiting strategies that don't mesh well with real human players. The paper proposes to use piKL, a prior search-based method that generates strong yet human-like plays, as the human policy to train the RL policies with. On the side of training the human policy, the method introduces (1) an iterative training scheme to improve piKL (the human policy) by feeding its successful play back to its dataset and (2) a way to learn human policies with different expertise levels by tuning the regularization strength. On the side of RL training with the human policy in-the-loop, to prevent RL overfitting to the human policy, the method proposes to further regularize the trained human policy by preferring “human-like” actions instead of greedily choosing the best move. Finally, the method also proposes to augment the RL policy with a search-based strategy to guard against catastrophic failure caused by out-of-distribution moves by real human players. The method is evaluated primarily via playing with real human players. The authors built a customized game server and solicited players of varying expertise level to play against the trained agent. The paper shows that the proposed method is able to outperform the RL policies trained with naive BC-based human policies.


**Summary Of The Review:**

Overall I quite like this paper. See my main comments for the remaining concerns.

---

> ### Author Response · Authors · 2022-11-12
> **Rebuttal**
>
> > Re: Does lambda truly capture the level of expertise in eq.2?
>
> The interpretation that lambda captures level of expertise is inferred from the original piKL paper where the authors show that piKL with certain lambdas achieves higher self-play performance while predicting the human moves on a test set better than the behavior clone policy trained to minimize the prediction loss, especially on games with high scores. Therefore, for a certain range of lambda, it is reasonable to assume that piKL models stronger human players. Smaller lambdas may behave more like unregularized optimal play. In this paper, the selected lambdas (1, 2, 5, 10) all have high prediction accuracy.
>
> > Re: The authors should discuss which aspect of the proposed method needs to be adapted in order to accommodate new domains.
>
> The general recipe of using behavioral clone regularized search as policy improvement operator in either IL or RL can be useful in many other domains that require collaborating and modeling. The underlying search algorithm can be swapped out. For example in a fully observable setting we may use MCTS. The real practical limitation is the availability of dataset, as well as a model/simulator so that search can be performed.
>
> > Re: suggested related work
>
> Thanks a lot for suggesting the missing related work. We will add more discussions on it in the related work section.

---

### Official Review · Reviewer_hc8j · 2022-10-25

**Confidence:** 2
**Correctness:** 2
**Technical Novelty And Significance:** 2
**Empirical Novelty And Significance:** 2
**Recommendation:** 3

**Clarity, Quality, Novelty And Reproducibility:**

I found that the paper lacked clarity regarding the overall objective and the methodology proposed. Without the introduction of piKL, it was extremely challenging to realize the different variants as piKL IL, piKL - LBS, piKL BR. I feel that I could have appreciated the work much better had it been more clear.

**Strength And Weaknesses:**

Strengths :
1. I agree to the problem statement of being of high importance with upcoming HiL methods where coordination with the human model / policy should be accounted for.

2. I like the fact that the work attempts to model humans of varying skill levels.

Weaknesses :

1. I was quite disappointed with the background section as the paper mentions piKL several times and admits to extend that work, but fails to provide any background details about this existing work.
2. Why Partial Observability is so important? Again, I feel with improved related work sections this could have been solved. The work mentions two key things in their setup - need for coordination with real humans as well as that agents exist in partially observable environments. Are there works that have already solved full observability?
3. The work keeps mentioning SPARTA several times in the related work section & some of its details - but not piKL? Is there a reason for it?
4. How feasible is such a technique in other environments (other than Hanabi) where 200k + simulations may not be available?


**Summary Of The Paper:**

The work extends piKL (an existing work for learning human-like policies) to coordinate with humans. They showcase their results on Hanabi benchmark.

**Summary Of The Review:**

I found that the work attempts to solve an interesting problem, relevant to the HiL community however lacks good readability to be accepted as a paper in ICLR. I would recommend the authors to update the write up to accordingly motivate the problem and highlight the contributions.

---

> ### Author Response · Authors · 2022-11-12
> **Rebuttal**
>
> > Clarity and overall recap:
>
> PiKL itself is a method that regularizes a search algorithm with human models to better model human players. It needs to be coupled with a search method. In Dec-POMDPs, one of the most powerful methods is SPARTA and its more generalized variant Learned-Belief Search (LBS). We use piKL with LBS as the underlying search algorithm in this paper. PiKL-IL is an iterative imitation learning method that alternates between generating better data with piKL-LBS and training a better policy on those generated data. piKL-BR incorporates the idea of piKL regularization into Q-learning to train a best response with RL.
>
> > Re: fails to provide any background details about piKL
>
> We discuss the piKL method in the last paragraph of Section 3.2 as part of the regularized search. We refer to it as “Jacob et al (2022)” The core mechanism of piKL that we employ is expressed in Equation 2. Apologies for not making it clearer. We will explicitly mention the piKL name in that paragraph.
>
>
> > Re: importance partial observability and commnet on related works
>
> Multi-Agent collaboration with partial observability is a widely studied problem in the AI community with many existing works and benchmarks advocating the importance of it. The partial observability makes coordination between multiple agents hard because agents need to infer the real intention of their partners (Theory of Mind reasoning). Since this paper does not propose a new problem setting but rather attempts to solve an important existing one, in related work we focus on comparing against prior works that address this problem.
>
>
> > Re: How feasible is such a technique in other environments (other than Hanabi) where 200k + simulations may not be available?
>
> Data: In this paper we focus on the scenarios where human behavior data are available but a learned behavioral clone policy is suboptimal and may lead to overfitting problems when used to model humans in real life. This setting is important and ubiquitous in AI research.
>
> Simulations: The need of the simulation is required by the underlying search algorithm and it may be eliminated once more advanced search techniques in Dec-POMDP are proposed. (For example, AlphaZero requires simulator because of MCTS, but later MuZero removes this requirement.)

---

> > ### Author Response · Authors · 2022-11-16
> > **Any additional comments?**
> >
> > Thank you again for your reviewing the paper as well as our rebuttal. We would like to know if our rebuttal has clarified the questions and whether there are additional questions that we can address before the end of the rebuttal period.

---

### Official Review · Reviewer_6xiK · 2022-10-26

**Confidence:** 2
**Correctness:** 4
**Technical Novelty And Significance:** 2
**Empirical Novelty And Significance:** 1
**Recommendation:** 3

**Clarity, Quality, Novelty And Reproducibility:**

I found the paper quite unclear to follow, with the Experiment details section particularly confusing. I was also confused about the  difference between the "2nd" and "3rd" stages of pikl3. There are too many acronyms/sub-methods (pikl-IL, pikl-LBC) that make the overall prose quite challenging to follow. I would encourage the authors to add more subsections and outlining to the experiments section.

**Strength And Weaknesses:**


While I believe multi-agent human coordination is a challenging and interesting problem, I overall found the paper quite difficult to follow.
My concerns include:

1. The gains of the proposed pikl3 method over baselines are not that strong at all, and is only evaluated on one task (Hanabi)
2. The methodological contribution largely seems to be via a regularization term, and applying piKL to multiple stages of the multi-agent task, which seems quite limited in novelty to me
Minor: \lambda  controls the regularization of unconstrained search towards human data, which is a bit different than different human "skill levels", as the spectrum is more about the degree of human-like behavior, making the overall point of different levels a bit confusing


**Summary Of The Paper:**

The paper considers the multi-agent human-AI collaboration setting, and the challenge of coordinating with humans. Specifically focusing on the Hanabi benchmark task, the authors propose the piKL3 method which as the following components:
(1) a human model that captures diverse skill levels controlled via a \lambda regularization term,
(2) train a human-like best response model,
and (3) further apply PiKL on the trained best-response model.
They evaluate their method on Hanabi, showing marginal improvement over a BC baseline.

**Summary Of The Review:**

Because I found the paper quite difficult to follow and limited in terms of novelty and performance of the proposed method, I lean towards reject. However, because of my unfamiliarity with the related works, this is a low confidence review.

---

> ### Author Response · Authors · 2022-11-12
> **Rebuttal**
>
> > Re: Clarity and overall recap
>
> Sorry for the confusion of many acronyms as this work indeed builds upon many prior works. The experiment section focuses on the detail on how the models are trained. The method section (Section 4) better explains how the three parts of the algorithm works individually and how we combine them together.
>
> In short, piKL is a prior method that proposes to regularize search towards a learned human anchor policy to better model strong human players. PiKL can be used with any search algorithm depending on the problem setting. In the Dec-POMDP setting considered in this paper, we use Learned Belief Search (LBS) as the underlying search method and hence the name piKL-LBS. PiKL-IL alternates between data generation with piKL-LBS and policy improvement on the generated data with imitation learning (spiritually similar to AlphaZero but replacing MCTS with piKL-LBS). PiKL-BR extends this idea to reinforcement learning to train a human-like best response against the output of PiKL-IL. Finally, at test time, we run piKL-LBS on top of piKL-BR to avoid obvious out-of-distribution errors.
>
>
> > Re: The gains of the proposed pikl3 method over baselines are not that strong at all, and is only evaluated on one task (Hanabi)
>
> The improvement of 1 point is significant in the context of Hanabi. As we point out in the paper, in reinforcement learning, improving a policy from 20 to 21 in Hanabi takes roughly the same amount of time as training a policy from 0 to 20. The difficulty of the Hanabi game grows superlinearly. To obtain one more point, the policy needs to be systematically more efficient and robust since any mistake in early game may cost many more losses later. Hanabi is one of the most challenging and popular partially observable multi-agent collaboration benchmarks. It has a large amount of hidden information and also requires complex strategic reasoning. There are hardly any other well-studied challenging benchmarks that satisfy the same criterias.
>
>
> > Re: The methodological contribution largely seems to be via a regularization term, and applying piKL to multiple stages of the multi-agent task, which seems quite limited in novelty to me.
>
> The core idea of the paper is to 1) use regularized search as a policy improvement operator (piKL-IL), 2) combine regularization with RL (piKL-BR) and then 3) further use regularized search to fix OOD errors at test time.
> The original piKL paper shows that piKL with proper lambda can predict the human behavior on the test dataset better than the behavioral clone policy trained solely for that purpose. Given that they predict human behavior well while improving the end-to-end performance in self-play, it is reasonable to infer that those lambdas model stronger human players.

---

> > ### Author Response · Authors · 2022-11-16
> > **Any additional comments?**
> >
> > Thank you again for your reviewing the paper as well as our rebuttal. We would like to know if our rebuttal has clarified the questions and whether there are additional questions that we can address before the end of the rebuttal period.

---

### Official Review · Reviewer_ywYA · 2022-10-27

**Confidence:** 4
**Correctness:** 3
**Technical Novelty And Significance:** 3
**Empirical Novelty And Significance:** 3
**Recommendation:** 5

**Clarity, Quality, Novelty And Reproducibility:**

The clarity and quality of the writing and experiments that are done seems to be quite good. I do think the evaluation is not comprehensive enough. The ways the authors use piKL also seems to be novel. In general there are enough experimental details for reproducibility.

**Strength And Weaknesses:**

The writing of the paper is quite good and the algorithm is described clearly. There are some nice algorithmic insights in the various pieces of the piKL3 algorithm. The empirical results in Table 3 also show a clear improvement over the baseline of BR-BC.

My main concern with the paper is that piKL3 is both complex and computationally expensive, and I'm not sure the limited evaluation justifies all the complexity. I understand that the human experiments are expensive to run, but it seems like there are a lot of other ways the authors could have done ablations or better evaluated whether all pieces of the algorithm are fully necessary. For instance, one could train a proxy agent using BC on some held-out data (possibly from a different population) not used for the algorithm. Ideally, there would be experiments showing what happens if you train piKL-BR directly on the BC policy rather than on the piKL-IL policy, or what happens if you don't add piKL on top of piKL-BR, etc. With the current evaluation, it's very hard to tell where the improvement over BR-BC is coming from—it could be a combination of all parts of the algorithm, or just one. The authors argue that "in practice, researchers may use any combination of the three components of piKL3 based on the properties and challenges of their specific domains." However, given how expensive some components are (most researchers in academia do not have access to 500 GPUs), it would be ideal if readers could get a better sense of which components are the most important and decide if the computational expense is worth it for each one.

Other specific issues:
 * In Table 1 I don't understand exactly how $\lambda$ can be varied when evaluating piKL-IL self-play. Isn't the output of piKL-IL, as you describe, a single neural network policy which ends up being the average of piKL policies with various values of $\lambda$? How can the $\lambda$ value of this be changed when evaluating it?
 * Where is footnote 3?

Typos:
 * Page 6: "whose input contains the mean of Gaussian distribution" -> "mean of Gaussian distributions"
 * Page 9: "all life token lost" -> "all life tokens lost"
 * Page 9: "the population of the human testers have a profound impact" -> "the population of the human testers has a profound impact"

**Summary Of The Paper:**

This paper looks to improve on the piKL method for producing good policies for human-AI collaboration. The proposed method, piKL3, uses piKL in three ways to produce a final collaborative policy. The first part, piKL-IL, is an iterative imitation learning algorithm which takes as initial input a behavior-cloned policy based on human data. At each iteration, the algorithm generates data sampled from piKL agents based on the current policy with various values of $\lambda$. Then, a new policy is trained via behavior cloning on the generated data and the process continues. The idea is to produce a policy with similar advantages to piKL but that better approximates a range of human skill levels. The second part of the algorithm, piKL-BR trains a best response to the output of piKL-IL with additional regularization that encourages the best response to be similar to the original behavior-cloned policy. Finally, the third part further improves the output of piKL-BR by using piKL again with the piKL-BR policy as both the anchor and rollout policy. Empirically, piKL3 seems to improve human-AI performance on Hanabi against a baseline of simply training a best-response to the behavior-cloned policy.

**Summary Of The Review:**

Overall, the paper is good except that it could use a better evaluation with ablations to justify the complex, multi-stage algorithm that is proposed. Therefore, I don't think it is quite ready for publication.

---

> ### Author Response · Authors · 2022-11-12
> **Rebuttal**
>
> > Re: main concern
>
> Given the high-variance nature of Hanabi, we would need at least 200 human games per method to further tell the difference between ablated variants. It is quite difficult to recruit enough players. The more games they need to play, the less people are willing to participate. Given that limitation, the paper aims to develop the best possible AI to coordinate with humans in zero-shot and compare that with human experts with convincing experiments. For that purpose, we have completed the largest-scale human experiments in Hanabi.
> The experiment suggested by the reviewer is a good idea. However, the dataset we currently use is merely sufficient to learn a decent human model and the performance of the behavioral cloning model drops too significantly to be useful if we split the dataset in half. For example, when we split out the validation and test set (about 2% of the total data), the self-play performance of the behavioral clone policy drops from 21.0 to 19.7.
>
> PiKL-IL with different lambda inputs performs significantly better than BC (21.81~22.66 vs 19.72). While piKL-BR trained against the BC policy may perform reasonably well with the general population in Table 2, it may not perform as well in Table 3 with more advanced players. Another benefit of using piKL-IL is that we can sample the input lambda explicitly to reduce overfitting. These points were discussed in the second paragraph of Section 4.2.
>
> In reality, which components are more important really depends on the task and the dataset quality. For example, in domains where human behavior data quality is worse, the piKL-IL step becomes more important. A detailed ablation in Hanabi may not transfer to other domains.
>
> > Re: In Table 1 I don't understand exactly how can be varied when evaluating piKL-IL self-play.
>
> The output is a single neural network but it can condition on different lambdas by taking lambda as input. Specifically, instead of conditioning on the actual real value of the lambda, we append a one-hot encoding representing the Gaussian “distribution” from which the lambda is sampled to the input of the neural network.
>
> > Re: footnote 3
>
> Sorry that is a typo. They should all be piKL3 instead of piKL^3

---

> > ### Author Response · Authors · 2022-11-16
> > **Any additional comments?**
> >
> > Thank you again for your time spent on reviewing the paper as well as our rebuttal. We would like to know if the reviewer has further comments that we can address before the end of the rebuttal period.

---

> > ### Comment · Reviewer_ywYA · 2022-11-22
> > **Thanks for the reply**
> >
> > Thank you for your detailed response to my comments. I think my overall impression of the paper remains the same: decent overall results, well written, and reproducible experiments, but the algorithm is complex, expensive, and has only limited evaluation.
> >
> > With respect to less data resulting in a worse BC model, wouldn't you still be able to show an improvement of piKL3 over BC-BR even if the BC models are worse? One could do an ablation reserving half the data for a human proxy and only compare against methods that use the same amount of data. Even if these methods perform worse than ones that use all the data, it would be a fair comparison and help to understand how much each part of the algorithm is contributing to the overall performance.
> >
> > I'm somewhat borderline on the paper—at least the results seem to be correct and well-explained, but it still seems like there is room for improvement in the evaluation. I don't think additional human experiments are feasible or necessary but I believe there are other ways the method could be more fully evaluated.

---

### Author Response · Authors · 2022-11-12
**Comment during rebuttal period**

We appreciate all the constructive feedback from the reviewers!

Please kindly go over our rebuttal and let us know if further clarification is needed.

---

### Decision · Program_Chairs · 2023-01-20

**Decision:**

Reject

**Justification For Why Not Higher Score:**

The paper has an algorithm which does ~1 point better than the previous methods. But there is not enough analysis to justify why we need such a complex algorithm for 1-point improvement. Hence I recommend reject.

**Justification For Why Not Lower Score:**

N/A

**Metareview: Summary, Strengths And Weaknesses:**

This paper introduces piKL3, a three-step algorithm that performs well while playing the game of Hanabi with an ad-hoc team of human players. The experimental results show that the algorithm has the best human-AI performance in the game of Hanabi.

One of the main criticisms of the paper is that the proposed algorithm is very complex and computationally intensive. While it shows performance gains, reviewer ywYA says that there is not enough analysis in the paper to justify different components of the algorithm. I would ignore other complaints which are all very minor or irrelevant. As reviewer ywYA says, while the paper is very interesting, it is not yet ready for publication. I encourage the authors to incorporate the feedback from the reviewers, add more extensive analysis and resubmit to a future venue.